# Analysis of High-Power Charging Limitations of a Battery in a Hybrid Railway System

**Mazhar Abbas** [1] **, Inho Cho** [2,*] **and Jonghoon Kim** [1,*]

1    Department of Electrical Engineering, Chungnam National University, Daejeon 34134, Korea; isphani111@gmail.com
2    Korea Railroad Research Institute, Uiwang-si, Gyeonggi-do 16105, Korea
\*    Correspondence: inhocho@krri.re.kr (I.C.); whdgns0422@cnu.ac.kr (J.K.)

**Abstract:** Fuel cell (FC)-driven railroad propulsion systems (RPSs) have been much appreciated for the past two decades to get rid exhausts of fossil fuels, but the inability of FCs to capture regenerative power produced by propulsion systems during regenerative braking and the dependency of its power density on operating current density necessitates the hybridization of FCs with batteries and/or supercapacitors to utilize the best features of all three power sources. Contrary to the research trend in hybridization where the purpose of hybridization such as fuel saving, high efficiency, or high mileage is achieved by certain operational algorithms without going into detail models, this study using detailed models explores the impact of high-power charging limitations of batteries on the optimization of hybridization, and proposes a solution accordingly. In this study, all three power sources were modeled, the optimal and suboptimal behaviors at the individual level were identified, and power distribution was implemented for a propulsion system, as recommended by the optimal features of all individual power sources. Since the detailed modeling of these power sources involves many mathematical equations and requires the implementation of continuous and discrete states, this study also demonstrates how, using C-MEX S-Functions, these models can be implemented with a reduced computational burden.

**Keywords:** railroad propulsion systems; power density; current density; state of charge; polarizations

## 1. Introduction

Fuel cell (FC)-driven railroad propulsion systems (RPSs) have emerged as proenvironmental conveyance systems because of the emission-free operation of FCs as power sources. Typical internal combustion engines (ICE) are 28%–30% efficient while electric motors achieve an efficiency of 85%–95%. Transmission losses in ICE-based drives are also higher as compared to losses in electrical transmission of power [1]. Early hybrid vehicles retain the diesel-based engines as the master engine and used batteries to let the ICE engines operate at the optimum range by compensating the extra loads that might cause ICE to deviate from an efficient operating range [2]. Although the development of batteries with the capability of propulsion increased the degree of electrification in the hybrid mix, it could not replace the ICE forever. The development of fuel-cells (FCs) made it possible to replace the ICE because the FC can give both high mileage and high power [3]. In addition, the directly fueled FC has a high tank-to-wheel efficiency compared with diesel engines. Most developed countries have already moved from combustion engines to FCs for powering locomotives [4]. FCs and batteries (Batt) as a primary source of power for propulsion have already become commercialized. Ultracapacitors (UCs) are also considered the best source for short bursts of power as they have a higher power density than FCs and Batt. Comparatively, FCs outperform Batt in refueling and UCs in energy density, whereas UCs and Batt have fast response as well as the capability to capture energy produced during regenerative

braking of propulsion systems. In applications where charge–discharge cycles are frequent, Batt life degrades fast, but UCs are least affected. Hybridization of power sources is intended to utilize the application-friendly characteristics of all power sources collectively [5].

In hybrid systems, depending on the load profile and the sizing of the power sources, the total cost of the system can be significantly reduced by the decision whether to supply all the load with one power source or the base power by one source with a supply of peak power by other sources. For smooth loads, a single source may perform at a high capacity factor, but a high disparity in peak and average power requirements results in the oversizing of the single source [6]. Even for smooth loads operated by such single sources as Batt or FC, it was estimated that the total cost, including capital, maintenance, fuel, and environmental costs, can be reduced if the operation is done in a hybrid way [7]. A simple configuration of a hybrid propulsion system is shown in Figure 1.

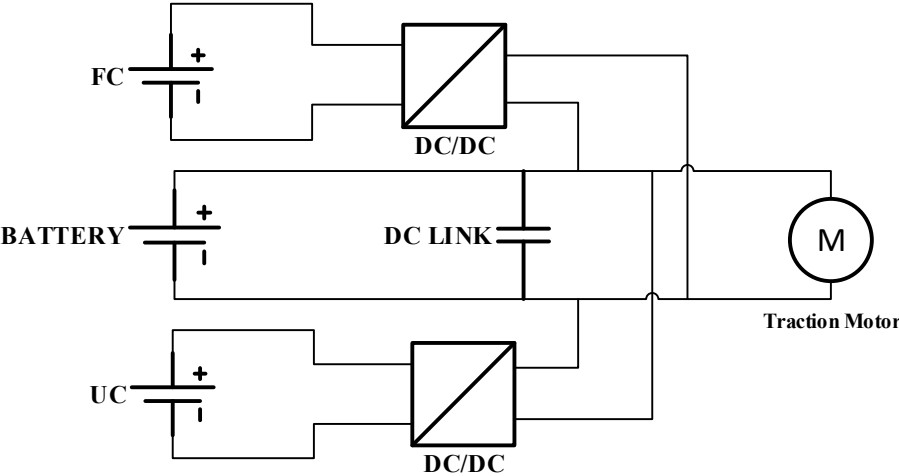

**Figure 1.** Fuel cell (FC) hybrid propulsion system configuration. Battery, FC, and UC are power sources of this hybrid system.

## 2. Literature Review

The primary objective of hybridization is optimization, but it is not necessary that random hybridization always gives optimized results. However, hybridization creates an option for optimization, and the exact algorithm for optimization is a function of targets and priorities. In the case of hybrid propulsion systems, hybridization has been studied with different contexts of optimization. For high-speed trains, the integration of renewable sources and hybrid energy storage systems was done in [8] to save energy and cost. In [9], an online model of FC capable of tracking the changing conditions with time, especially aging, was developed, and using this model, an optimal power splitting between FC and battery was devised to find the best efficiency and power operating points for FC. The hydrogen consumption minimization strategy implemented for the FC hybrid system in [10] showed that it can enhance drivability and economy. The energy management algorithm developed in [11] intended to mitigate the impact of high-frequency load on FC. These studies first identified the constraints such as low efficiency, high fuel consumption, high degradation of life, and accordingly, proposed solutions. However, these studies did not mention the issues associated with the sizing of the power sources.

Regenerative braking gives back energy to the system which is stored in the battery/supercapacitor. The energy from regenerative braking reduces the sizing of the system. The sizing of the hybrid system was done in [12] considering the uncertainty in regenerative braking. The life degradation of Li-ion batteries is very sensitive to deep discharge. In hybrid systems, the rating of a battery just in terms of maximum energy demand does not avoid the risk of deep discharge. On the other hand, an increase in the sizing of a battery to avoid deep discharge increases the mass and volume of the system. The sizing of a battery was investigated in [13] to identify an appropriate balance between the system mass, system volume, and the deep discharge. The load pattern is one of the primary determinants in the

sizing. Based on the load pattern, the power sources of the hybrid system are combined, different sizes for each source are tried, and the corresponding capital cost and operational cost are calculated. Such a study has been carried out in [14] to minimize the total system cost, which includes the capital cost and operational cost. The impact of maximum C-rate on the system weight and fuel consumption was studied in [15]. The results show that a higher C-rate reduces both the system weight and fuel consumption. Although plenty of work has been done about the sizing of batteries, there are few studies about the impact of the limitations in high-power charging of a battery on sizing. One of the motivations of this study is to explore the consequences of high-power charging limitations of a battery on the sizing of the system, and subsequently on the optimization.

Secondly, some studies about hybrid propulsion systems did not consider UCs as an option; rather, they were limited to FCs and Batt only. These studies did not discuss the capturing of regenerative power [9,16]. In hybridized systems, assuming power sources as simple voltage sources, power flow optimization is possible, but power resource management, source operation at the optimal point, and sizing optimization need a detailed model of every source. Many studies have formulated novel power flow algorithms, but least discussed is the behavior of sources in response to the algorithms [10,17]. Briefly, it can be inferred that optimizations applied in hybrid propulsion systems without detailed models of sources cannot cover all aspects of optimization. The second motivation of this study is to signify the full model-based optimization by modeling all the power sources, identifying the optimal features of every source, and combining all the sources to get the best results from all the sources.

The scientific goal of this study is to highlight the high-power charging limitations of the battery, its undesirable consequences, and optimal solution to avoid the consequences. The battery suffers from under-sizing during discharge due to lacking the capability of high-power charging, and under-sizing makes the battery unreliable. Oversizing of the battery or hybridization with a capacitor can address the issue of limited high-power charging. Oversizing is not a cost-effective method while hybridization with a capacitor is an optimal method.

In Section 3, the model development is briefly discussed. Based on the models, sources are characterized, and the optimal range of operation is identified for every source. In Section 4, the power distribution among sources is defined based on both the merits and demerits of the sources highlighted in Section 3, as well as the traction load taken from the actual train platform. Section 5 concludes the paper, followed by suggestions for future work.

## 3. Methods

### 3.1. S-Function: Alternative to Bulky Modeling

The detailed modeling of the battery, fuel cell, and supercapacitor involve many mathematical equations. Modeling of these power sources was implemented in Simulink by [11]. In [12], the power sources were modeled in PSIM. The MATLAB coding was used to implement the power sources in [18]. For simulation purposes, these methods are appropriate. However, for real-time implementation, C-coding is commonly used. C-MEX S-Functions are programmed in C/C++, and also compensate for Simulink. Large mathematical models written in C++ can be easily linked to Simulink in the form of a rectangular box projecting out just input and output lines. An FC model implemented using C-MEX S-Function is shown in Figure 2. The parameters on the left side of Figure 2 are the inputs of the model and parameters on the right side are the outputs. All other mathematical equations are embedded inside a single rectangular block captioned as S-Function. The working of the model is expressed by the Equations (1) and (2). The terminal voltage of FC is a function of oxygen pressure at the cathode, total pressure at the cathode, hydrogen pressure at the anode, load current, working temperature, and water content in the PEM membrane.

$$V\_fc \; = \; Eopen - (V\_act + V\_ohm + V\_conc) \tag{1}$$

$$V\_fc \; = \; f\left(P_{ca}, P_{O2\_ca}, P_{H2\_an}, I_{FC}, T_{st}, \lambda\right) \tag{2}$$

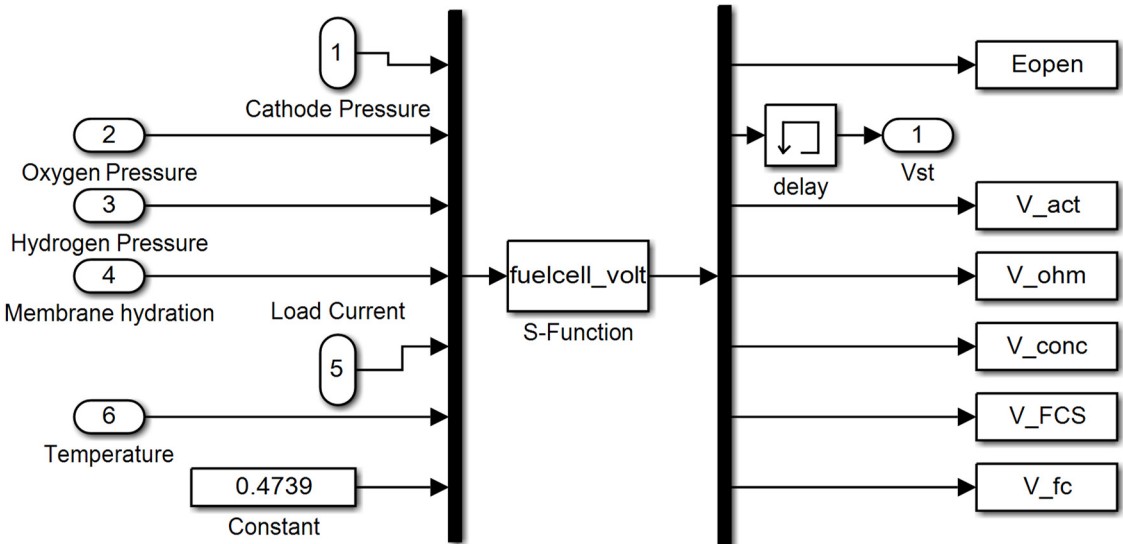

**Figure 2.** Display of FC model developed using S-Function.

Additionally, the modeling of power sources involves an update of states and integrals. It has built-in functions to store the states and execute the integrals without using extensive codes. For instance, the equations for estimating the state of charge of the battery is shown in Equation (3). The second part of the equation on the right side is just the accumulation of current with respect to time. Each time, the current accumulation is updated and the following value of SOC is obtained. The update of the current accumulation is automatically done in C-MEX S-FUNCTION if it is defined as a state. The implementation of an update with normal Simulink involves loops and delays, may raise the error of discontinuity, and also take more computation time.

$$SOC[k+1] = SOC[k] - \frac{\Delta t}{Cap}\eta[k]i[k] \tag{3}$$

There are many equations governing the modeling of power sources that involve states. As a reference, some of the equations are shown in Equations (4)–(6). Equation (4) is the coulomb counting equation for state estimation of the battery. Equation (5) models the terminal voltage of the supercapacitor. Equation (6) relates the pressure at the cathode of the FC to the mass of oxygen. The states in these equations and the update method are explained to signify C-MEX S-Function for modeling. $x[0]$ was modeled as a discrete state as it is the simple integration of current, and it was updated on every default call. For discrete states, the update commands are defined by the function "MDL_UPDATE". $x[1]$, $x[2]$, and $x[3]$ were modeled as a continuous state, and the values were calculated by the derivative function of S-Function to quantify and update the voltage, transient voltage, and reactant mass, respectively, on every default call.

$$SOC = 1 - \left[\left\{\left(\frac{1-SOC_0}{100}\right) \cdot Cap\right\} - \frac{1}{Cap}x[0]\right] \quad if\ x[0] = \int_{t_0}^{t_1} I_{batt}dt \tag{4}$$

$$V_{uc} = R_{epr} \cdot \left(\frac{x[1]+x[2]+I_{uc} \cdot R_{esr}}{R_{epr}+R_{esr}}\right)$$
$$if\ \frac{d}{dt}(x[1]) = \left(\frac{I_{uc}}{C_s}\right) - \left(\frac{\frac{V_{uc}}{C_s}}{R_{epr}}\right),\ \frac{d}{dt}(x[2]) = \left(\frac{I_{uc}}{C_0}\right) - \left(\frac{\frac{V_{uc}}{C_0}}{R_{epr}}\right) - \left(\frac{\frac{x[1]}{R_s}}{C_0}\right) \tag{5}$$

$$P_{O_2\_ca} = x[3] \cdot R_{O_2\_ca} \cdot T_{st}$$
$$if\ \frac{d}{dt}(x[3]) = \frac{dm_{O_2}}{dt} = W_{O_2\_ca(in)} - \left(W_{O_2\_ca(out)} + W_{O_2\_reacted}\right) \tag{6}$$

*x*[1] is voltage and the equation linked to this state is the rate of change of voltage. The function "MDL_DERIVATIVES" calculates the voltage from the linked equation. The syntax of the update and derivative call are given in Appendix A.

### 3.2. FC Modeling

Propulsion systems need power sources with the capability of quick startup and frequent on/off, and, among different types of FC, PEMFC's low-temperature operating range makes it well suited to cope with these demands of propulsion systems [19]. At the primary level, voltage and current give enough information about the dynamics of the FC, but there are multiple underlying factors affecting the voltage and current, such as temperature, pressure, stoichiometry, and flow of reactants. These factors drive the dynamics of FCs in automotive applications, as discussed elsewhere [20], and there are a number of other variables that need to be controlled. Further, the significance of these variables varies from application to application. In addition to dynamic behavior, nonlinearity may also occur because of improper hydration of the membrane. Nonlinearity projects a reflection in the output voltage in the form of bi-stability and oscillations. Thus, humidity control becomes critical to avoid undesirable nonlinear behavior [21]. To identify the optimal operating range and verify the performance of FC against the real data, a comprehensive model with the reflection of multi-physics behavior is required [22]. In this work, a model has been developed based on the laws of thermodynamics, kinetics, and electrochemistry already deduced in previous work [23]. The mathematical equations describing FCs have been coded in C language and linked to Simulink by C-MEX S-Functions.

The dynamics of FC in any application reflect in its terminal voltage $V_t$. The current magnitude in general, and waveform in particular, can be seen as the representative blueprint of the application. The current drawn gives shape to the curve of $V_t$ by directly introducing the polarizations or changing other factors inside the FC, which in turn impacts the FC voltage by adding polarizations, or in both ways. Thus, to capture the impact of the application on the FC dynamics, the $V_t$ waveform has been simulated as a function of current in Figure 3. It is simulated as a function of other factors that are current-dependent, such as oxygen pressure and cathode pressure in Figures 4 and 5, respectively. The current density at which the FC is operated also determines the power density, as shown in Figure 6.

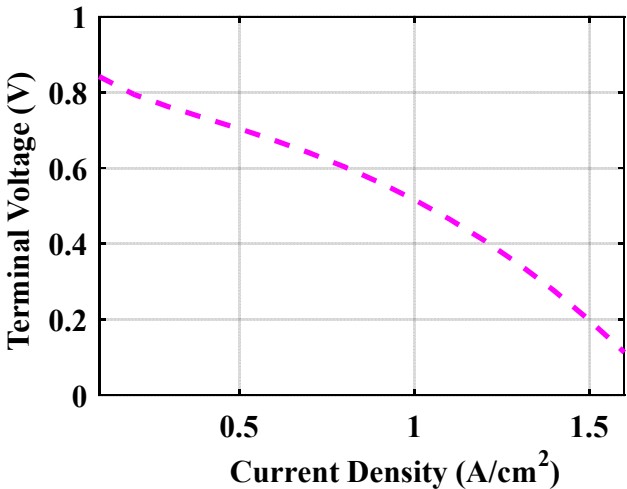

**Figure 3.** FC terminal voltage as a function of current density.

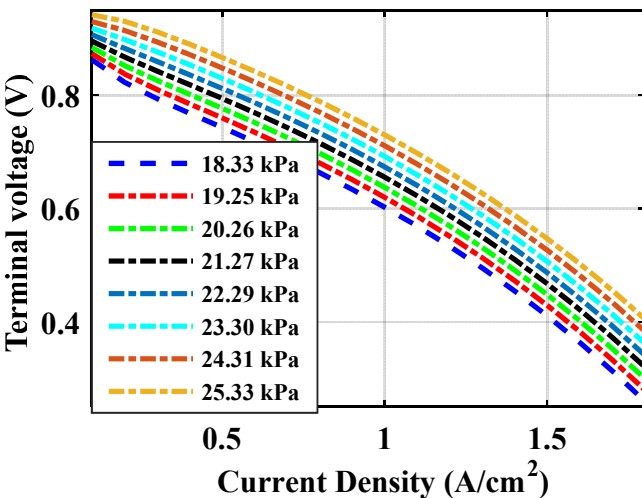

**Figure 4.** FC terminal voltage as a function of oxygen pressure at the cathode.

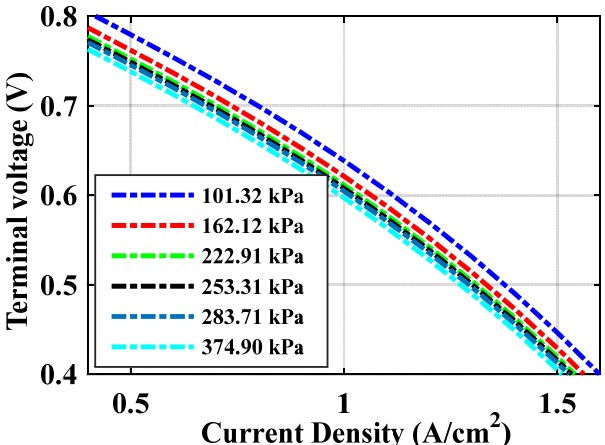

**Figure 5.** FC terminal voltage as a function of total pressure at the cathode.

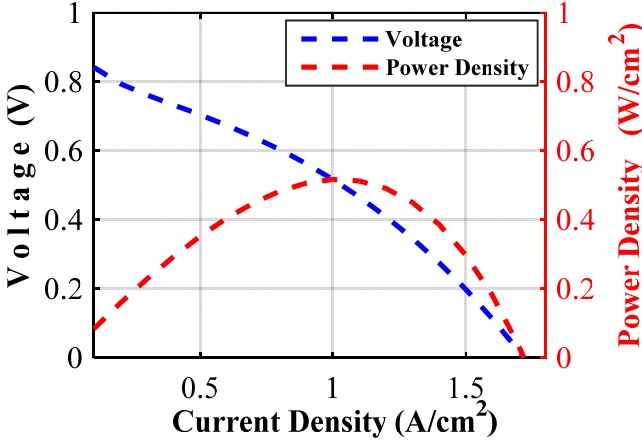

**Figure 6.** FC power density as a function of current density.

It can be inferred from Figures 3 and 4 that the voltage provided by the FC will change upon changing the load current. To provide the constant voltage, the pressure of oxygen and cathode must be maintained at a proper reference. The FC controller can regulate the voltage supplied to the compressor in such a way that the reference pressure of oxygen and the cathode will be maintained. Second, the application of FCs in RPSs is the result of the higher power density for acceleration, and

the power density of the FC becomes maximum at a certain range of current density, as shown in Figure 6. Thus, the current density has been set as another control variable to operate the FC at a high power density under all conditions of the load. Thus, the optimums to be implemented are:

i.　　　Voltage regulation under changing load;
ii.　　　Operation of FC at high power density.

Because the high power density is available at a certain small range of current density, operating the FC at this small range least affects the voltage. Thus, the intention to operate the FC at a high power density automatically compensates for the voltage regulation controllers.

### 3.3. Capacitor Modeling

A mathematical model introduced in previous research [24] was used to study the characteristic behavior of UCs. The mathematical equations have been given already in Section 3.1. The state of charge (SOC) and voltage of the UC at high power with the same charge and discharge time are shown in Figures 7 and 8, respectively. Figure 7 shows that the SOC decrease at discharging is the same as the SOC increase at charging because high-power charging is possible in UCs, and this feature has significance in RPSs. The battery can be sized either considering the maximum charging rate or discharge rate. However, Figure 8 shows that the voltage decrease in UCs is almost proportional to the SOC decrease, so the voltage decreases abruptly in response to the decrease in SOC, and this characteristic is a demerit of UCs.

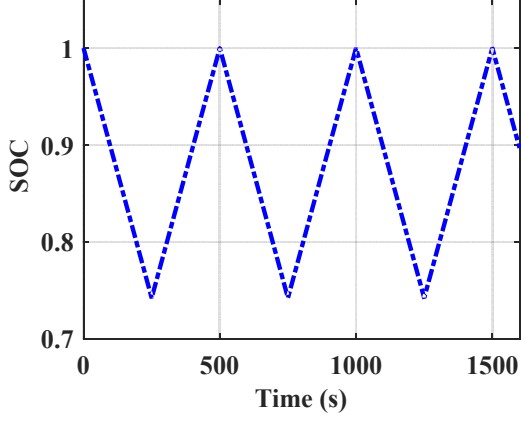

**Figure 7.** Change in state of charge (SOC) of ultracapacitor (UC) during charging/discharging.

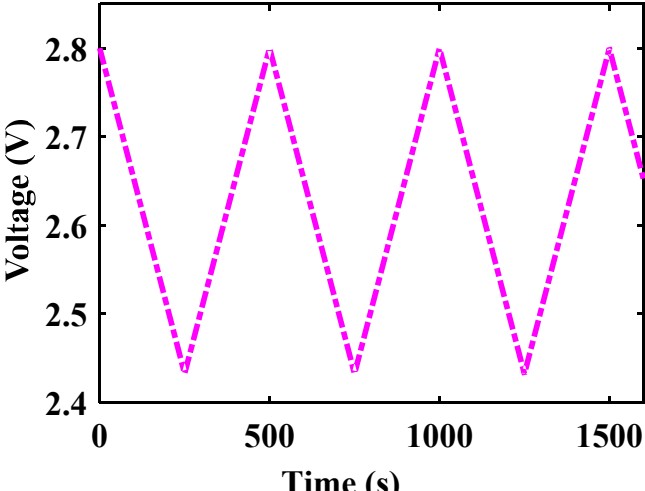

**Figure 8.** Change in terminal voltage of UC during charging/discharging.

### 3.4. Battery Modeling

In battery modeling, the relation between SOC and open-circuit voltage (OCV) was studied from experimental results. The OCV was calculated for SOC, ranging from 100% to 10% SOC using the experimental pattern described in Figure 9; then, using the curve-fitting tool in MATLAB, the equation that relates SOC to OCV, was extracted. A pulse discharge is applied to the battery for 10 s at a certain SOC, and the battery is then allowed to rest for 1 h. The battery retains to equilibrium position during rest time and the voltage at equilibrium is the open-circuit voltage. The equation for SOC estimation was already discussed in Section 3.1. The polarizations were calculated using the mathematical model developed in previous work [25]. Considering the specifications of the battery INR-21700-40T, for low-power, voltage and SOC were simulated with the same charge and discharge time, as shown in Figure 10. Then, for high power, SOC has been simulated with the same charge and discharge times, as shown in Figure 11. This figure shows that the SOC decrease at discharging is different from the SOC increase at charging because of the high-power charging limitations of the battery, and the charging limitations add more cost to Batt in RPSs.

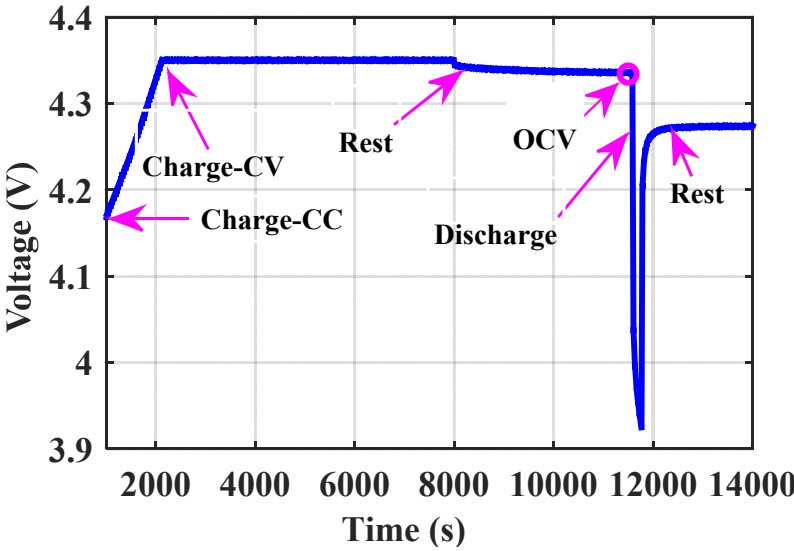

**Figure 9.** Experimental method for estimation of open-circuit voltage.

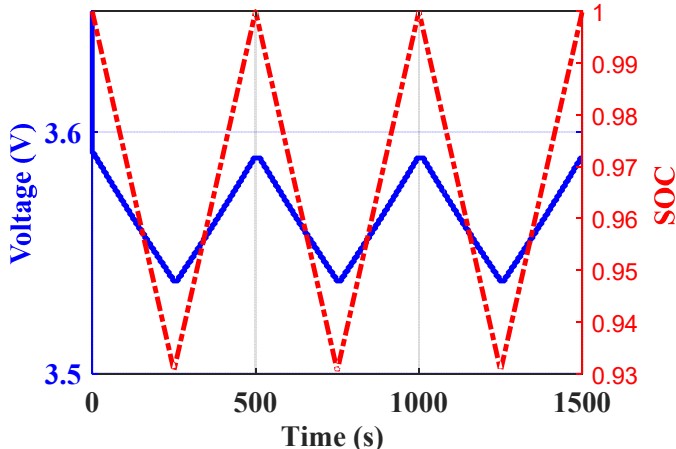

**Figure 10.** Trend matching between the terminal voltage and SOC of the battery.

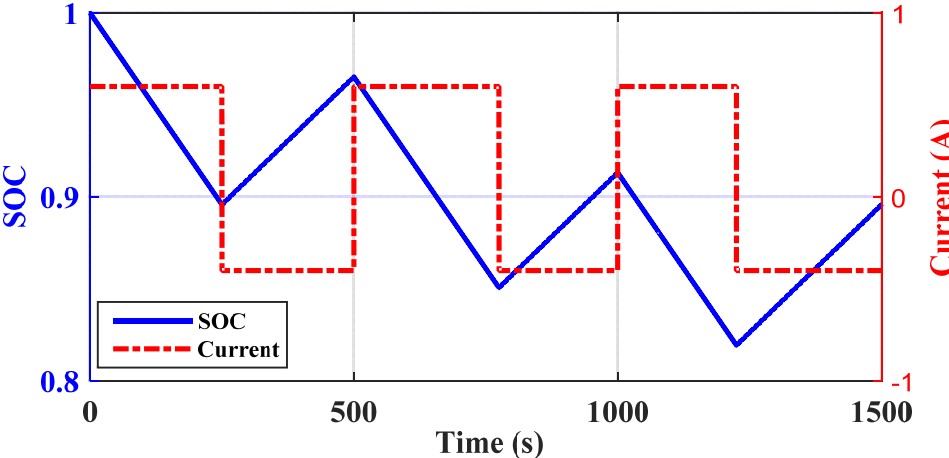

**Figure 11.** Change in SOC considering limitation in high power charging.

### 3.5. Load and Locomotive Modeling

The actual power demand profile used in this study (Figure 12) is a reflection of both the load and locomotive model. The traction power $P_w$ required at wheel is a function of driving speed $v$, air drag coefficient $c_x$, rolling coefficient $f_r$, mass of the propulsion system $M$, equivalent mass of the rotating parts $m_j$, gravity constant $g$, air density $\rho$, and frontal surface area $A$ as expressed in Equation (7). The power demand $P_d$ at the terminal of power sources is greater than the traction power due to losses in the converter, inverter, gear, and motor that transmit power from power sources to the traction wheel. Thus, the efficiency of electric motor $\eta_{em}$, inverter $\eta_{inv}$, converter $\eta_{conv}$, and gear $\eta_{gear}$ impact the power demand. The power demand relates to traction power as Equation (8).

$$P_w = v\left[\frac{1}{2}\rho A c_x v^2 + Mg f_r + (M + m_j)\frac{dv}{dt}\right] \tag{7}$$

$$P_d = \max\left\{\frac{1}{\eta_{em}}\frac{1}{\eta_{inv}}\frac{1}{\eta_{conv}}\frac{1}{\eta_{gear}}P_w\right\} \tag{8}$$

The power demand has been calculated by taking into account the values of all the parameters discussed above, so the load profile qualitatively reflects the intended locomotive model. The values of some of the key parameters of the intended locomotive were included in Table A1 of Appendix A.

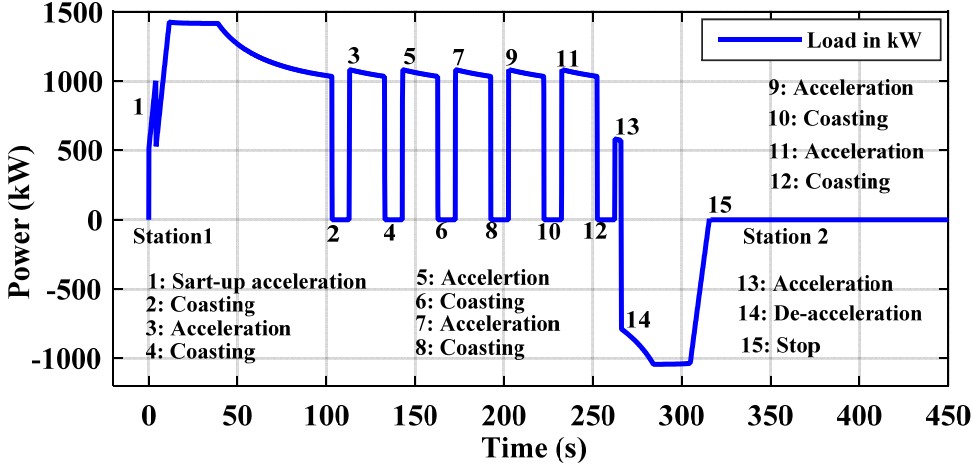

**Figure 12.** Train's actual load profile between two stations (Emulated).

### 3.6. Load Distribution between Battery and FC

High mileage and fast refueling of FC make it an appropriate choice to supply base load. For the same mileage as obtained by FC, the required size of the battery is two times the size of FC in terms of weight and volume. On the other hand, the battery responds faster to the frequently changing peak loads as compared to FC. The FC takes about 103 to 104 s to reach a steady state after a change in load resistance, while the battery response time is in milliseconds. The fast response of the battery makes it well suited for the supply of peak loads. Considering all the aspects explored above, the distribution of load is determined as under also is shown in Figure 13.

i.　　Since FC has a high energy density, the base load is provided by the FC, and it was sized to provide an average load of approximately 500 kW plus the auxiliary load of 80 kW.

ii.　　The operating point of the FC lies in the range of high power density, and, accordingly, the FC was configured as shown in Table 1.

iii.　　During no load, the FC energy is used for charging the Batt or UC.

iv.　　The peak power is provided either by Batt or UC.

v.　　The energy consumed during discharging of Batt/UC should be regained during charging at the end of every station to maintain operational reliability.

vi.　　All the energy produced during regenerative braking must be captured by Batt/UC for economical operation.

For the defined patterns of charging and discharging, the size of the battery should be enough to store energy in order to supply the load demand without interruption. Hybridization of the UC and battery with FC are evaluated as two separate cases to show the incentives of high-power charging capability of UC and disadvantages of limited high-power charging of the battery.

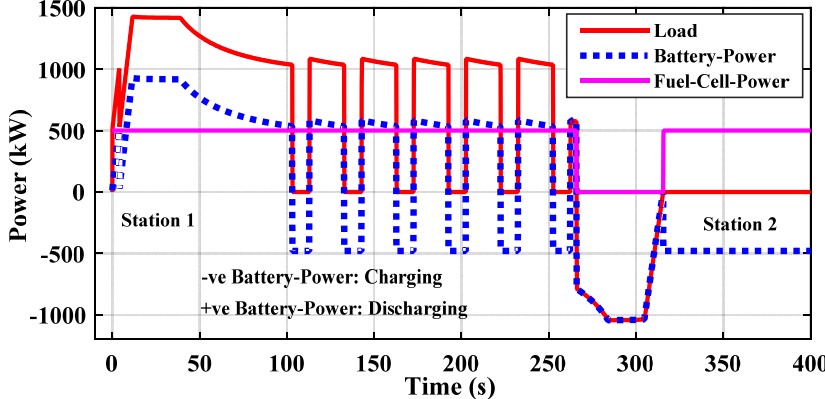

**Figure 13.** Load power distribution of actual load profile among power sources.

**Table 1.** Configuration and specification of FC for hybridization with a battery.

| Cell | PEMFC | Number of Modules | 2 in Series |
|---|---|---|---|
| Company | DOOSAN Korea | Stacks in each module | 5 in parallel |
| Cell operating voltage | 0.5 V | Total number of stacks | 10 |
| Cell operating current | 0.85~0.9 A/cm$^2$ | Cells in each stack | 480 in series |
| FC pack power | 580 kW | Total cells | 4800 |
| FC pack voltage | 480 V | FC stack voltage | 240 V |
| FC pack current | 1225 A | FC stack current | 225 A |
| FC active area | 250 cm$^2$ | Power density | 4.5 W/cm$^2$ |

## 4. Consequences of High-Power Charging Limitations of the Battery

*Results and Analysis*

The battery is sized considering the recommended discharge rate by the manufacturer and the load demand. The power rating of the battery is kept higher than the maximum power of the load. For reliability of the load supply, the battery should retain the same SOC at the end of the station equal to that at the start of the station. The cells are connected in series to provide the voltage level required by the load.

For load distribution as in Section 3.6, at first, the battery pack was used to supply peak load. The battery pack was rated for maximum load by considering the specifications of one of the high-power cylindrical cells INR-21700-40T shown in Table 2. The results of the battery SOC are simulated in Figure 14. It shows that the net SOC decreased after every station, and, after some stations, its SOC reaches zero. There are two types of consequences caused by such behavior of SOC. The battery life is a function of SOC. It is not recommended to operate the Li-ion battery at lower SOC because of the faster decay of life at lower SOC. Secondly, the drop of SOC to zero results in the interruption of supply to load, and such unexpected failure of power supply decreases the reliability of the hybrid system.

**Table 2.** Configuration and specification of the battery for hybridization with FC.

| Cell | INR 21700 40T | Battery Pack Rated Current | 1050 A |
|---|---|---|---|
| Company | Samsung SDI | Number of modules | 5 in parallel |
| Capacity | 4000 mAh | Stacks in each module | 6 in parallel |
| Nominal voltage | 3.6 V | Total number of stacks | 30 |
| Rated discharge per cell | 35 A | Cells in each stack | 300 in series |
| Rated charge per cell | 6 A | Total cells | 9000 |
| Battery pack power | 1134 kW | Discharging per cell | @ 30 A |
| Battery pack voltage | 1080 V | Charging per cell | @ 5 A |

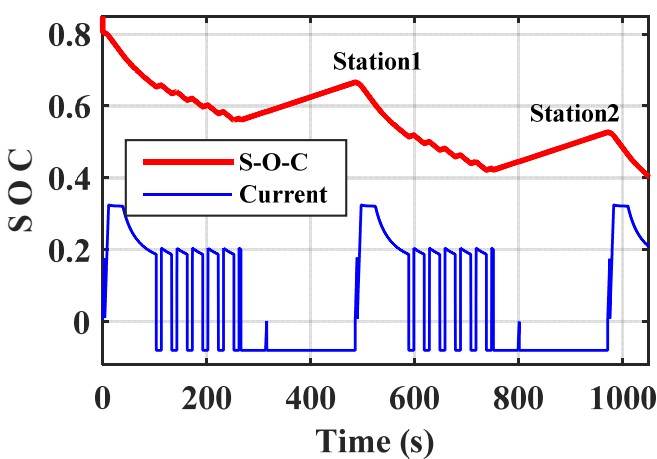

**Figure 14.** Battery's state of charge with actual load profile scaled (sized for maximum load).

The battery could also capture only a small amount of the regenerative power (Figure 15a) and the remaining regenerative power will be wasted. Both these issues are caused by the charging limitations of the battery. The battery discharges at high power but is unable to get the charge back due to the limitations of high-power charging. To compensate for this issue, the battery was rated for five times the maximum load. Now, the SOC reached the initial point after every station (Figure 16), and also the battery could capture more of the regenerative power (Figure 15b).

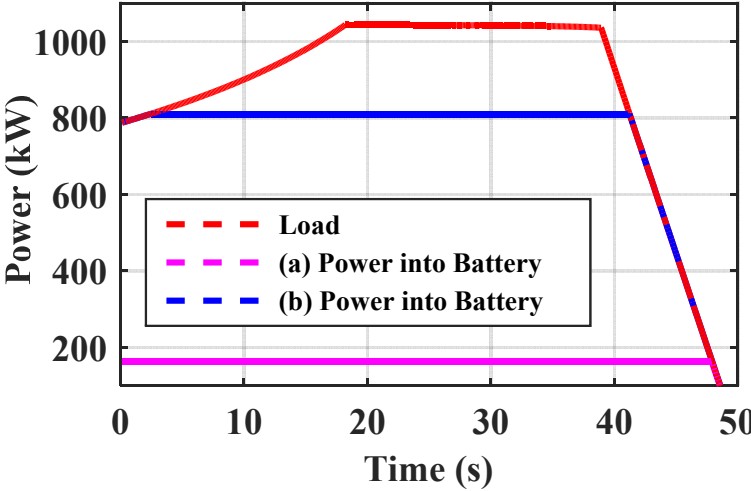

**Figure 15.** Regenerative power capture by batteries of different sizing (**a**). Rated for maximum load (**b**) Rated for five times of maximum load.

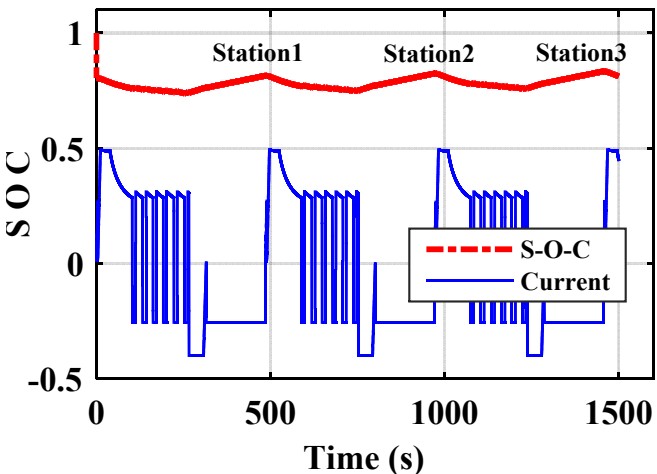

**Figure 16.** Battery pack's state of charge with actual load profile scaled (sized for five times of maximum load).

Then, the UC pack was rated for maximum load by considering the specifications of the LS Mtron shown in Table 3. The results for SOC and regenerative power-capturing are shown in Figures 17 and 18, respectively. The results for the UC pack rated for maximum load are better than those of the Batt pack rated five times higher than the maximum load. The SOC was returned well to the initial value after every station, and also the UC could capture more regenerative power compared with Batt. Based on results produced by the high-rated battery and UC, it is stated that the solution to consequences caused by the high-power charging limitations of the battery can be either the overrating of the battery or the hybridization of the UC with the battery or the replacement the UC with the battery. The oversizing is a non-optimal solution, and besides high cost, it will increase the mass and volume of the hybrid system. The use of UC with the battery is a better option. The UC cannot be used alone because there are some issues associated with the UC, such as low energy density, rapid voltage decay, and short supply time at high power.

The UC can provide short bursts of high power for 30 s, but the period for the startup acceleration of the train is approximately 110 s. Keeping in view these demerits, for the said application, both the battery and capacitor should be used. Using UC in collaboration with Batt will not only reduce the maximum power rating requirement of the Batt, but it will also increase the regenerative power-capturing capacity of the overall propulsion system.

**Table 3.** Configuration and specification of UC for hybridization with FC.

| Item | LSUC 002R8S0100FEA | UC Pack Rated Current | 1110 |
| --- | --- | --- | --- |
| Company | LS Mtron | Number of modules | 3 in parallel |
| Capacitance | 100F | Stacks in each module | 5 in parallel |
| Nominal voltage | 2.7 V | Total number of stacks | 15 |
| Rated current per cell | 74 A | Cells in each stack | 386 in series |
| Operating current per cell | @ 60 A | UC pack voltage | 1080 V |
| UC pack power | 1198 kW | Total number of cells | 5775 |

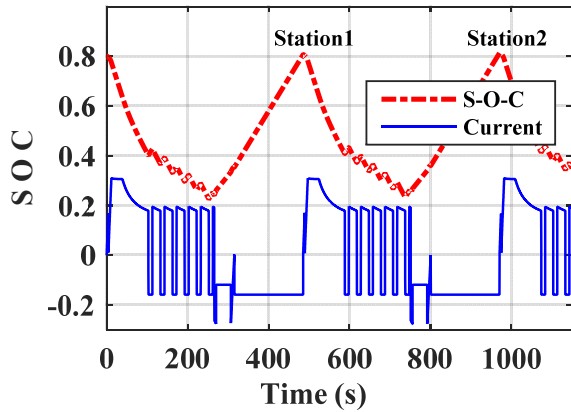

**Figure 17.** Ultracapacitor state of charge with actual load profile scaled.

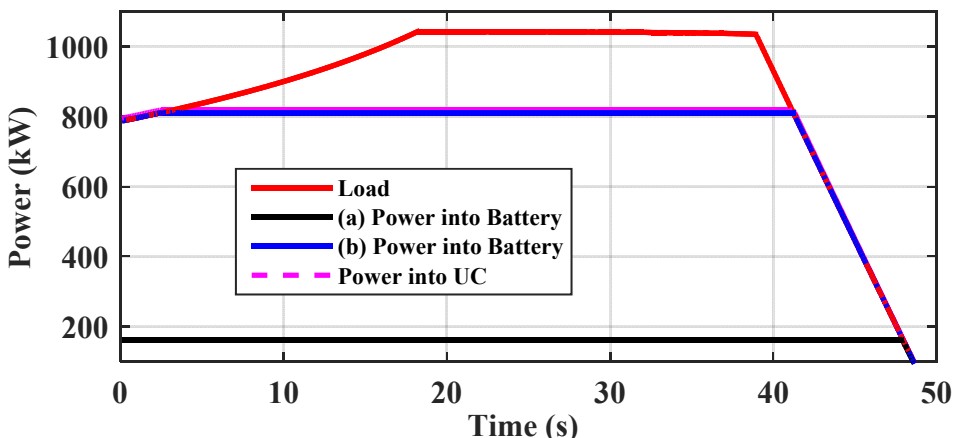

**Figure 18.** Comparison for regenerative power capture between UC and battery.

The analysis of results is summarized as follows. Operating FC within the range of high power density along with a supply of base load compensates for all FC issues discussed in Section 2 and, more importantly, the FC has a good capacity factor throughout the operation. For peak load supply, it was shown that UC outperformed the battery for capturing regenerative power, and also the SOC could be well-retained in the UC after every station. The low energy density and high self-discharge of UC suggests the hybridization with a battery for better results.

## 5. Conclusions

Mathematical models of the Batt, UC, and FC were implemented using C-MEX S-FUNCTION to highlight its significance in modeling. Based on the models, optimal and suboptimal features of these power sources were identified with respect to applications in RPSs. The significance of the optimal

features of all sources in RPS application was discussed based on the simulated and emulated results. As recommended by the selective characterization, load distribution among the power sources was defined. Following the load distribution, the battery was sized for maximum power requirements, and tested by applying the load profile. The consequences of the high-power charging limitations of the battery were observed. The battery was overrated and tested again. It was observed that overrating can compensate for the high-power charging limitations. Then, the UC was sized for maximum power and tested by applying load. The results were compared with that of the battery. The results suggest that undesirable consequences caused by the high-power charging limitations of the battery can be avoided either by the overrating of the battery or by the hybridization of the UC with the battery or by replacing the UC with the battery. The oversizing is a non-optimal solution, and also the UC alone cannot satisfy the load demand. It is recommended that both Batt and UC should be used in collaboration to supply the peak load to improve the operational reliability and maximize the energy capturing from regenerative braking.

The current study considers only one type of battery cell for hybridization. It is the chemical characteristic of currently existing batteries that a high-power battery cannot provide high energy and vice versa. However, the load may demand both high power and high energy. This study can be further extended by considering hybrid types of battery packs. In the hybrid type, both the high power and high energy cells can be used together to compensate for uncertain requirements of power and energy by load.

**Author Contributions:** M.A. contributed to the main idea of this article, and wrote the paper. I.C. provided the experimental data. J.K. revised the paper critically. All authors have read and agreed to the published version of the manuscript.

**Funding:** This research was funded by a grant (20RTRP-B146008-03): "Development of Optimized Railway Vehicle Propulsion System using Hydrogen Fuel Cell Hybrid Power System (1.2 MW or more)" from the Ministry of Land, Infrastructure and Transport, South Korea.

**Acknowledgments:** This research acknowledges a grant (20RTRP-B146008-03): "Development of Optimized Railway Vehicle Propulsion System using Hydrogen Fuel Cell Hybrid Power System (1.2 MW or more)" from the Ministry of Land, Infrastructure and Transport, South Korea. It also acknowledges Korea Railroad Research Institute for research collaboration and provision of data.

**Conflicts of Interest:** The authors declare no conflicts of interest.

*Abbreviations*

| | | | |
|---|---|---|---|
| RPSs | Railroad propulsion systems | DC | Direct current |
| ICE | Internal combustion engine | CV | Constant voltage |
| FC | Fuel cell | CC | Constant current |
| UC | Ultracapacitor | PEM | Polymer electrolyte membrane |
| kPa | Kilopascal | $f$ | Function of |

*Parameters*

| | | | |
|---|---|---|---|
| $E_{open}$ | Open-circuit voltage (V) | $R_s$ | Series resistance ($\Omega$) |
| $V_{\_st}$ | Stack voltage (V) | $I_{uc}$ | Ultracapacitor current (A) |
| $V_{\_act}$ | Activation loss (V) | $C_s$ | Capacitance (F) |
| $V_{\_ohmic}$ | Ohmic loss (V) | $C_0$ | Transient capacitance (F) |
| $V_{\_conc}$ | Concentration loss (V) | $P_{O2\_ca}$ | Oxygen pressure at the cathode (kPa) |
| $V_{\_fc}$ | Fuel cell voltage (V) | $P_{ca}$ | Pressure at the cathode (kPa) |
| $V_{\_FCS}$ | Fuel cell system voltage (V) | $P_{H2\_an}$ | Hydrogen pressure at anode |
| SOC | State of charge (Unit less) | $\lambda$ | PEM Membrane hydration (Unit less) |
| $SOC_0$ | Initial state of charge (Unit less) | $T_{st}$ | Stack temperature (°C) |
| $Q$ | Nominal capacity (Ah) | $I_{FC}$ | Fuel cell current (A) |
| $I_{batt}$ | Battery current (A) | $m_{O2}$ | Mass of oxygen (kg) |
| OCV | Open-circuit voltage (V) | $W_{O2\text{-}ca\_in}$ | Flow rate of oxygen into cathode |
| $V_{uc}$ | Voltage of ultracapacitor (V) | $W_{O2\text{-}ca\_out}$ | Flow rate of oxygen out of cathode |
| $R_{esr}$ | Equivalent series resistance ($\Omega$) | $W_{O2\text{-}ca\_reacted}$ | Flow rate of reacting oxygen |
| $R_{epr}$ | Equivalent parallel resistance ($\Omega$) | $R_{O2\_ca}$ | Gas constant for oxygen |

## Appendix A

Code Syntax

```
# define MDL_UPDATE
static void mdlUpdate( SimStruct *S, int_T tid)
{
real_T *x = ssGetRealDiscStates(S);
inputRealPtrsType uPtrs = ssGetInputPortRealSignalPtrs(S, 0);
```
$$x[0] = \tfrac{1}{Cap} \times \int I dt$$
```
}
# define MDL_DERIVATIVES
static void mdlDerivatives(SimStruct *S)
{
real_T *dx = ssGetdX(S);
real_T *x = ssGetContStates(S);
inputRealPtrsType uPtrs = ssGetInputPortRealSignalPtrs(S, 0);
```
$$dx[1] = \left(\tfrac{I_{uc}}{C_s}\right) - \left(\tfrac{V_{uc}/C_s}{R_{epr}}\right),$$
$$dx[2] = \left(\tfrac{I_{uc}}{C_0}\right) - \left(\tfrac{V_{uc}/C_0}{R_{epr}}\right) - \left(\tfrac{x[1]/R_s}{C_0}\right),$$
$$dx[3] = W_{O_2\_ca(in)} - \left(W_{O_2\_ca(out)} + W_{O_2\_reacted}\right)$$
```
}
# define MDL_DERIVATIVES
static void mdlDerivatives(SimStruct *S)
{
real_T *dx = ssGetdX(S);
real_T *x = ssGetContStates(S);
inputRealPtrsType uPtrs = ssGetInputPortRealSignalPtrs(S, 0);
```
$$dx[1] = \left(\tfrac{I_{uc}}{C_s}\right) - \left(\tfrac{V_{uc}/C_s}{R_{epr}}\right),$$
$$dx[2] = \left(\tfrac{I_{uc}}{C_0}\right) - \left(\tfrac{V_{uc}/C_0}{R_{epr}}\right) - \left(\tfrac{x[1]/R_s}{C_0}\right),$$
$$dx[3] = W_{O_2\_ca(in)} - \left(W_{O_2\_ca(out)} + W_{O_2\_reacted}\right)$$
```
}
```

**Table A1.** Details about load modeling.

| Operation Speed [km/h] | 110–121 |
|---|---|
| Weight [ton] | 128 |
| Maximum force [kN] | 121.833 |
| Inverter efficiency [%] | 98 |
| Traction motor efficiency [%] | 96 |
| DC/DC converter efficiency [%] | 96 |
| Gear efficiency [%] | 92 |

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
