# Peer review of "Analysis of High-Power Charging Limitations of a Battery in a Hybrid Railway System"

_electronics, doi:10.3390/electronics9020212_

Round 1

Reviewer 1 Report

The authors addressed some limitations of high-power charging of a battery in a hybrid railway system.  C MEX S‐Functions have been used to implement the detailed modelling of the power sources (Batt, UC, and FC).  The paper has been written well, however some comments and suggestions are detailed below:

1-      Previous studies in implementing the power sources need to be highlighted comparing with the impact of C MEX S‐Functions.

2-      Implemented “easily” in abstract is vague, please clarify more. Is it based on CPU time or what?

3-      Model development Section 2, is the developed model (Eq 1 to Eq 4) new? I think some references are needed.

4-      I suggest add a notations section for your parameters and variables that have been used in the models.

Reviewer 2 Report

The paper tackles the analyses of applicability combined power sources for railway systems. The authors analysed the system containing a fuel cell, a battery and an ultracapacitor. The topic fits the scope of the journal well. The paper might have value for science and general practice, however I have a number of remarks which should be taken into account and incorporated in the paper before publication.

Main issues:

The scientific goal of the paper is not clearly highlighted. In the introduction alternatives to the presented system should be mentioned at least, e.g. a diesel-electric systems or application of mechanical support for presented system (flywheel, hydraulic/pneumatic accumulators etc.). Presentation of research methodology is insufficient. Please provide a clear information on assumptions and research plan, inputs and outputs of the used model as well as information on the range of measured parameters. Please provide description of all the symbols used in equations (1)-(4). Provide information on measuring units as well. Figure 2 – please provide description of all the modules/symbols used in the model. The Conclusions should be updated with information on the possibility of further development of the presented system.

Technical and formal issues:

Page 5 – I suggest to present the code of the functions in the Appendices. Figures 5-6 – ( A/cm2 ) -> (A/cm2). Figures 7-18 – ( sec ) -> (s). Figures 4, 5 and 9 – please provide description of symbols. Figure 5 – I suggest to use SI measure units. Please recalculate atm into kPa or MPa. Figures 5-18 – please remove upper captions. Table 2 – please move it below the reference of this table in the main text. Tables 1-3 – please improve the captions. What “approach” is taken into account? Figure 18 – please do not finish the section with a figure. Provide an additional discussion. List of all symbols and abbreviations used in the paper would improve the quality of the communication.

Reviewer 3 Report

Topic of the paper is interesting and very important. Autroes prepared enough literature review, in my opinion.

Figures have good quality and are well described in the text.

One comment to the structure of the paper - it would be better to divide first chapter and seperate introduction and literature review.

Aurhors should also add plans of further research in the topic (in Conclusion chapter).

Round 2

Reviewer 2 Report

The authors have addressed most of my remarks. The paper is much better now. Thank you. I have only some minor remarks regarding notations. After incorporating these changes I recommend the paper to be published.

Minor remarks:

Eq. (2) - I suggest to change a type of brackets: {} into (). Eq. (4) - please add integration limits. Eq. (4),(5), (6) - please change a cross product into a dot product (x ->  ⋅ ). Eq. (6) - please change slashes into horizontal division lines (/ -> -). Eq. (7) - I suggest to increase height of square brackets to be the same like in Eq. (4). Fig. 6 - Watt -> W.
